# Crowding in or crowding out? How local government debt influences corporate innovation for China

**Junbing Xu[1], Yuanyuan Li[2], Dawei Feng[2]\*, Zhouyi Wu[2]\*, Yang He[3]**

**1** Wang Yanan Institute for Studies in Economics (WISE), Xiamen University, Xiamen, China, **2** Institute of Industrial Economics, Jiangxi University of Finance & Economics, Nanchang, China, **3** School of Finance, Jiangxi University of Finance & Economics, Nanchang, China

\* fengdawei@jxufe.edu.cn (DF); Zhouyi90@hotmail.com (ZW)

**Data Availability Statement:** The data underlying the results presented in the study are available from the Csmar database and the Wind database. If readers want to use the metadata, they can visit the website: https://www.gtarsc.com/ and https://www.wind.com.cn/NewSite/edb.html.

## Abstract

The pressure upon local governments to redeem their debt could affect government fiscal ability. It could consequently affect their fiscal policies on corporations, which might distort corporate innovation. Based on the data of Chinese Shanghai and Shenzhen A-share listed companies and the local government implicit short-term debt financed by local government financing vehicles (LGFVs) in 31 provinces, this paper shows that local government debt (LGD) negatively affects corporate R&D investment in China, thereby suggesting a strong crowding-out effect. The crowding-out effect is more pronounced when the firm is a non-state-owned enterprise (NSOE), the firm's size is small, the firm's age is young, or the firm is in the lower market competition. This paper provide evidence by interacting the terms that local government actions, such as consumption of fiscal resources, strengthening tax collection efforts, or consumption of credit resources, might partially account for the crowding-out effect. This study illustrates the innovation costs of local government debt.

## 1. Introduction

Since the 2008 financial crisis and the 2010 European debt crisis, the economic consequences of government debt, especially the effects of government debt on micro-enterprises, have attracted the attention of academia and government regulators alike. Although some studies have examined the impact of government debt on corporate debt maturity [1–3], corporate investment [4,5], and corporate leverage [6,7], the manner in which government debt affects firms' innovation decisions is still not clear. This study fills this gap by focusing on the effect of local government debt (LGD) on corporate innovation in China.

The LGD discussed in this paper is the local government implicit short-term debt financed by local government financing vehicles (LGFVs) in 31 provinces and must be redeemed within one year. In China, the LGD (LGFVs' short-term liabilities) increased from 2.9 trillion yuan in 2012 to 8.76 trillion yuan in 2018. The principal and interest debt redemption pressure brought by LGFVs' debt is testing the local government's fiscal ability [8,9], which would shape the local government's relevant fiscal, taxation, and credit policies for enterprises.

**Funding:** The author(s) received no specific funding for this work.

**Competing interests:** The authors have declared that no competing interests exist.

However, innovation, being a risky asset [10], often requires these policies from the government. Therefore, from a theoretical perspective, local government debt will have an impact on corporate innovation. In this study, we explored three channels by which LGD affects corporate R&D investment. That is, (1) the LGD will consume the regional fiscal resources available to enterprises, which would reduce corporate R&D investment [11,12]; (2) the LGD will strengthen the local government collection efforts on the corporation, which would reduce corporate R&D investment [13–16]; and (3) the LGD will consume regional credit resources available to enterprises, which would reduce corporate R&D investment [17–20].

Based on the data of Chinese Shanghai and Shenzhen A-share listed companies and LGFVs from 2012 to 2018, we empirically tested the impact of LGD on corporate R&D investment from the perspective of LGD redemption pressure. We found that LGD has significantly crowded out corporate R&D investment. In terms of economic significance, with every standard deviation increase in local government debt, the proportion of corporate R&D investment decreases by 0.037 to the mean. The crowding-out effect is more pronounced when the firm is a non-state-owned enterprise (NSOE), the firm's size is small, the firm's age is young, or the firm is in the lower market competition. Further, we provide suggestive evidence that consuming fiscal resources, strengthening tax collection efforts, or consuming credit resources might partly account for the crowding-out effect.

Our results are robust because we conducted a series of robustness checks as follows: (1) using the sum of LGFVs' bond issuance amount with the same maturity as the instrumental variable, (2) alternative measures of the corporate R&D investment and local government debt, (3) changing the empirical model, (4) excluding the interference of other fiscal and taxation policies, and (5) employing other robustness checks.

Our study contributes to the following aspects. First, it adds to the literature on the micro-economic consequences of government debt on corporate. Research has explored how government debt affects corporate debt maturity [1–3], corporate investment [4,5], and corporate leverage [6,7]. Meanwhile, our study focuses on the effect of LGD on corporate innovation in China. Second, our study enriches the literature on the determinants of corporate innovation, especially on how LGD shapes corporate R&D investment choices. The literature on the determinants of corporate innovation is mainly based on internal corporate [21–23] and external corporate [24,25] factors. In this study, we explore the determinants of corporate innovation from the perspective of local government debt. Finally, our study improves the understanding of LGD in China from a government debt redemption pressure perspective. Research has estimated the total LGD with no geographical breakdown [26] or (the studies) based only on bond insurance, which accounts for a small part of total debt by LGFVs [6]. In contrast, we build a detailed dataset on total short-term debt by LGFVs in 31 provinces between 2012 and 2018. We use a unique data to study the economic consequences of LGD on corporate innovation from a debt redemption pressure perspective.

The remainder of the paper is organized as follows. Section 2 introduces China's institutional background. Section 3 shows the literature review and theoretical hypothesis. Section 4 describes the empirical design. Section 5 depicts the main estimation results, and Section 6 for further analysis. Section 7 reports robust checks, and conclusions are exhibited in Section 8.

## 2. Institutional background

Since 1978, with the reform and opening policy, China has achieved rapid economic growth by relying on cheap labor and investment in infrastructure construction [27]. However, with the recent rise in labor costs, this growth model is considered outdated [28]. The country's top leaders are promoting innovation as the key to sustained economic growth [25]. Further, the

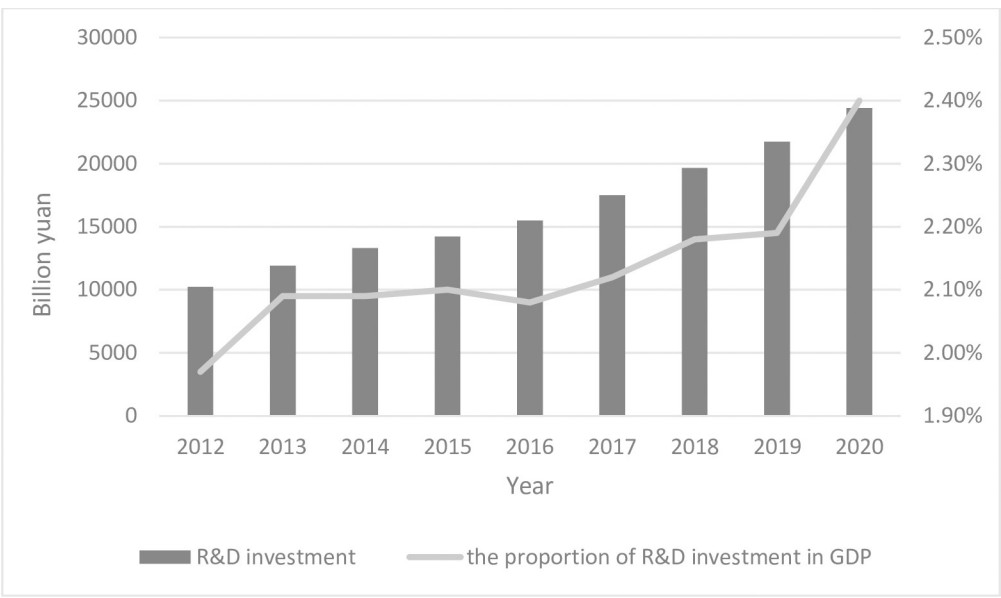

**Fig 1. Chinese government innovation investment.**

country's R&D investment has risen from 10,240 billion yuan in 2012 to 24,426 billion yuan in 2020. As shown in Fig 1, the proportion of R&D investment in GDP has been increasing yearly.

In China's "13th Five-Year Plan (The Five-Year Plan for national economic and social development is one of the most important plans of the government. It records the national strategy during this period)," innovation is regarded as the primary guiding principle of the economic policy. Since strong economic intervention and command characteristics exist in the country, local governments have actively adopted various fiscal policies, such as fiscal subsidies, transfer payments, loan policy, and tax incentives, to support innovative activities. Specifically, unlike some developed countries, owing to China's imperfect financial system and intellectual patent protection laws, the government's fiscal policy for corporate innovation activities becomes more important.

China's financial system is underdeveloped [29]. It comprises the banking and intermediation sector, financial markets, shadow financial sector, and foreign sectors [30]. Compared with other developed countries and emerging economies, China's financial system has always been dominated by a large banking system [30]. Bank loans remain the most important financing channel for innovation in the country. However, banks are more inclined to provide loans to large enterprises, especially state-owned enterprises (SOE) [31], as they believe that this type of enterprise has less risk of repayment and can obtain strong government assistance [32]. Conversely, most Small and medium enterprises (SMEs) and non-state-owned enterprises (NSOE) are difficult to obtain funds for innovation activities from banks. Although China's stock and bond markets are established, the conditions for companies to go public and requirements for issuing bonds remain quite strict. Thus, most corporates cannot obtain funds from the financial market. The shadow financial and foreign sectors are not sufficiently large to provide enterprises with adequate and continuous funding for innovative activities, and their costs are generally relatively high. Therefore, the government's fiscal policy is essential to address the financing constraints of corporate innovation activities.

China's laws on the protection of intellectual property rights are weaker than that of some developed countries [25]. Thus, the corporates' innovation benefits are difficult to guarantee.

Owing to the high cost and uncertainty of the benefits of innovation, corporates have no incentive to carry out such activities. The Chinese government cannot ameliorate its financial system and patent protection laws in the short term. Thus, it must adopt some fiscal policies to encourage enterprises to carry out innovative activities to maintain sustained economic growth.

## 3. Literature review and theoretical hypothesis

### 3.1 Literature review

This paper is mainly related to two aspects of the literature: (1) influencing factors of corporate innovation and (2) economic consequences of government debt.

**3.1.1 Relevant literature on the economic consequences of government debt.** The normal question in the study of macroeconomic consequences of government debt is how government debt affects economic growth. Classical economic scholars believe that government debt is not conducive to economic growth. They argue that, because of the low credit risk of government debt, government debt crowds out private investment [33–35]. However, Keynesian scholars believe that governments can rely on their reputation and raise funds from the market (e.g., through bonds), which will then be invested in public projects. Consequently, the public projects will improve basic public services, thereby promoting economic growth [36,37]. Other scholars have studied the economic and social consequences of government debt from the perspectives of household consumption [38] and corruption [39].

In the past, most of the literature on the economic consequences of government debt was analyzed from a macro perspective. However, an increasing volume of research has begun to focus on the impact of government debt on corporates. For example, Greenwood et al. [1] proposed the "gap-filling theory" of corporate credit debt maturity selection (GHS). They attested that enterprises will choose long-term bonds with lower costs when the government's long-term debt supply shrinks. Lugo and Piccillo [3] extended the one-country model into a two-country model based on the GHS (2010) model. They held that the debt maturity structure of enterprises is not only affected by the issuance of bonds by the government in their home country but also by the issuance of bonds by relevant governments of other countries. Relying on the GHS (2010) model, Van Bekkum et al. [40] investigated the reference interest rate of sovereign debt, and concluded that there is an alternative between corporate debt and government debt. This alternative will be weakened if sovereign debt fails to provide a high-quality reference interest rate. Badoer and James [2] established that a gap-filling effect exists owing to the impact of long-term government bonds on long-term bonds of high-credit companies. Eidam [41] investigated the public debt of Eurozone countries and averred that this gap-filling effect is more significant in government(s) with low financial constraints and high credit. Huang et al. [4,5] used corporate data from 69 countries to study public debt and found that public debt squeezes corporate investment by tightening corporate credit constraints.

**3.1.2 Relevant literature on the influencing factors of corporate innovation.** Currently, academic research on the influencing factors of enterprise innovation is based on internal and external perspectives. From an internal perspective, Barker and Mueller [42] believe that young executives with engineering backgrounds will provide a company a unique innovation advantage. Lin et al. [43] assert that managers' incentive mechanisms can promote the innovation activities of private enterprises. Benmelech and Frydman [44] find that CEOs with military backgrounds will lead to lower corporate R&D spending. Islam and Zein [23] hold that an inventor CEO will promote enterprise innovation by providing professional identification and transferring incentive credit to internal individual inventors.

From an external perspective, Benfratello et al. [24] believe that the relaxation of bank regulation would encourage more credit funds to flow into product or process innovation projects, which could significantly improve R&D quantity and quality. Acharya and Subramanian [45] argue that, when bankruptcy law is favorable to creditors, it causes excessive liquidation, reducing innovations in organizations. Fang et al. [25] contend that intellectual property rights protection can stimulate enterprise innovation.

## 3.2 Theoretical hypothesis

Corporate innovation is a long-term investment with high risk, which requires financial support from the government and banks. However, because of the large number of Municipal Bonds and LGFVs' bank debts, to redeem their debt, local governments may be prompted to consume regional fiscal resources, strengthen tax collection efforts on corporations (to increase fiscal revenue), or consume bank credit resources. In this study, considering these three possible actions of local governments, we analyze how LGD affects corporate R&D investment behavior.

First, an increase in LGD will reduce the regional fiscal resources available to non-urban investment corporations. Specifically, when the LGDs (LGFVs' short-term debts) become larger, to avoid debt defaults, local governments have to choose to repay the due debt first, which, in turn, consumes the regional limited fiscal resources. Thus, it will reduce the fiscal resources to non-urban investment enterprises. As mentioned earlier, the government's fiscal subsidies provide the incentives for enterprises to carry out innovative activities, as well as the funds they need. With the weakening of financial subsidies, corporates will directly face rising innovation costs and reduced profits, further leading to a reduction in innovation activities and R&D expenditures [46]. Specifically, the reduction of government fiscal subsidies may indirectly send a negative message to the market [47]. This situation will greatly dampen the innovative spirit of the enterprise and is not conducive to innovation development. Notably, existing literature also generally argues that a reduction in fiscal resource subsidies will reduce corporate R&D investment [11,12]. Therefore, the greater the LGD in a region, the more the fiscal resources that will be consumed, which would finally crowd out fiscal subsidies for corporate's R&D expenditure, and is thus not conducive to innovation.

Second, an increase in LGD will prompt local governments to strengthen their tax collection efforts on corporations which, in turn, would increase the corporate tax burden. This tendency is because, when LGD increases, their fiscal expenditure also rises as they must spend on debt repayments. Currently, the Chinese fiscal system is facing the pressure of decreasing revenue and increasing expenditure, and the fiscal operation is in a state of "tight balance". Therefore, the fiscal expenditure brought by LGD further increases local government fiscal pressure. To alleviate this fiscal pressure, local governments may be prompted to strengthen their tax collection efforts, which will increase the tax burden of non-urban investment corporations. This move will, in turn, crowd out corporate R&D investment [13–16]. Moreover, tax burden reduces the corporates' cash flow, worsens the corporates' financial constraints, and therefore, reduces, the investment level in innovative activities [48,49].

Third, an increase in LGD will reduce the regional credit resources available to non-urban investment corporations. Specifically, when the LGDs (LGFVs' short-term debts) become larger, to avoid debt defaults, local governments may coordinate with local financial institutions to continue lending to them, which consumes limited regional credit resources. Finally, it will reduce the credit resources to non-urban investment enterprises. Notably, existing literature argues that a reduction in credit resources will reduce corporate R&D investment [17–20]. Specifically, in China, where the financial system is underdeveloped, bank loans are the

most important external financing method for corporates to obtain funds for innovative activities. The local government's excessive use of regional credit resources has worsens the financing constraints of enterprises and further reduced the level of investment of enterprises in innovation activities [17]. Therefore, the greater the LGD in a region, the more the credit resources that will be consumed, which would finally crowd out non-urban investment corporation's R&D investment.

Based on these three scenarios, we propose the following main research hypotheses:

Hypothesis 1: *Holding everything else constant, the greater the local government debt, the lower the corporate R&D investment.*

## 4. Empirical design

### 4.1. Sample selection

Our study selected Chinese Shanghai and Shenzhen A-share listed companies from 2012 to 2018 as the primary sample. Following common practices in corporate finance and research requirements, we cleaned the data as follows. First, we excluded firms with abnormal financial or other conditions (ST firms), firms with risk of delisting stocks (ST* firms) and suspended and delisted firms. Second, we excluded firms with missing dependent and control variables. Third, we excluded listed local government financing vehicle corporations. Firm-level fundamental financial data were obtained from the China Securities Market and Accounting Research database.

The data of LGFVs were obtained from the Wind database of Municipal Bonds issued between 2008 and 2018. The liabilities of local governments were calculated through financial statements. Other provincial macro data were from the Wind database of the macroeconomic sector. Following common practices in corporate finance, all firm-level continuous variables were winsorized at the top and bottom 1%. The final sample consisted of 14314 firm-year observations.

### 4.2 Empirical design

We designed the benchmark model as follows:

$$RD_{p,i,t} = \alpha_0 + \beta LGD_{p,t-1} + \gamma Control_{p,i,t} + \delta_t + \theta_p + \tau_{ind} + \varepsilon_{p,i,t} \qquad (1)$$

where $p$ is the province, $i$ is the firm, and $t$ is the time. Explained variable $RD_{p,i,t}$ is the ratio of R&D investment to assets in firm $i$, provinces $p$, and in year $t$. $LGD_{p,t-1}$ is the core explanatory variable of the model, namely, the local government debt. The specific calculation is the ratio of the sum of short-term liabilities of all local government financing vehicles in province $p$ in year $t$−1 (short-term borrowing + notes payable + one-year current liabilities + other current liabilities + short-term bonds payable) to the fiscal expenditure in province $p$ in year $t$. We choose the LGFVs' short-term liabilities in year $t$−1 because these short-term liabilities are redeemed in the next year $t$. The real fiscal pressure on the government is in year $t$, which affects corporate behavior in year $t$. We choose the short-term liabilities of LGFVs rather than the total liabilities (long-term liabilities + short-term liabilities) because local governments are unlikely to default long-term liabilities in the short term and therefore their settlement will not affect current local government behavior. We also use fiscal expenditure to deflate the local government debt.

*Control*$_{p,i,t}$ represent the control variables in the model including provincial-level control variables (i.e., LnGDP and LnPop) and corporate level control variables (i.e., SIZE, SOE, AGE,

**Table 1. Variable definitions.**

| Variable Name | Abbreviations | Definitions |
|---|---|---|
| Explained variable | RD | Firm R&D investment on assets, R&D investment/total assets |
| Explanatory variable | LGD | Local government debt, the sum of short-term liabilities of all local government financing vehicles in province $p$ in year $t-1$/the fiscal expenditure in province $p$ in year $t$. |
| Control variable | LnGDP | Ln(GDP) |
| | LnPop | Ln(Population) |
| | SIZE | Firm size, Ln(total assets) |
| | SOE | An indicator variable that equals one if the stock is a Chinese state-owned enterprise, and zero, otherwise. |
| | AGE | Firm age, The current year + 1 -The establishing year |
| | LEV | Firm leverage, Total debt/Total assets. |
| | ROA | Return on assets, Net profit/Total assets. |
| | FIX | Fixed assets on assets, Fixed assets/Total assets. |
| | INT | Intangible assets on assets, Intangible assets/Total assets. |
| | SHR1 | Ownership concentration, the fraction of shares held by the largest shareholder. |

LEV, ROA, FIX, INV, and SHR1). The model also includes the province fixed effect ($\theta_p$), industry fixed effect ($\tau_{ind}$), and year fixed effect ($\delta_t$). Further, $\varepsilon_{p,i,t}$ is the random error term. Table 1 provides detailed definitions of all variables.

## 4.3 Summary statistics

We report the summary statistics of the main variables in Table 2. The mean value of the core explained variable, RD, is 0.022, which indicates that corporate R&D investment is relatively low in China. The mean value of the core explanatory variable, LGD, is 0.374, thereby indicating that the LGD pressure is high. The minimum value of LGD is 0.001, and the maximum value is 2.371, thus indicating that there are differences in LGD pressures between provinces in China.

**Table 2. Summary statistics of the main variables.**

| Variables | Num. | Mean | Std. | Min | Max |
|---|---|---|---|---|---|
| | | Panel A: Province level Variables | | | |
| RD | 14314 | 0.022 | 0.018 | 0 | 0.098 |
| LGD | 217 | 0.374 | 0.404 | 0.001 | 2.371 |
| LnGDP | 217 | 28.140 | 0.968 | 24.970 | 29.910 |
| LnPop | 217 | 8.127 | 0.840 | 5.729 | 9.337 |
| | | Panel B: Firm level Variables | | | |
| SIZE | 14314 | 22.02 | 1.236 | 19.97 | 26.02 |
| SOE | 14314 | 0.285 | 0.452 | 0 | 1 |
| AGE | 14314 | 2.863 | 0.294 | 2.079 | 3.555 |
| LEV | 14314 | 0.391 | 0.198 | 0.050 | 0.862 |
| ROA | 14314 | 0.047 | 0.058 | -0.183 | 0.215 |
| FIX | 14314 | 0.210 | 0.144 | 0.005 | 0.646 |
| INV | 14314 | 0.045 | 0.040 | 0 | 0.230 |
| SHR1 | 14314 | 34.55 | 14.43 | 8.735 | 73.33 |

**Notes:** Variable definitions are provided in Table 1.

**Table 3. Local government debt and corporate R&D investment.**

| Variables | Dependent variable: RD | | |
|:---:|:---:|:---:|:---:|
| | **(1)** | **(2)** | **(3)** |
| LGD | -0.002*** | -0.002*** | -0.002*** |
| | (-3.589) | (-3.234) | (-4.129) |
| LnGDP | | 0.004** | 0.004** |
| | | (2.431) | (2.493) |
| LnPop | | -0.008 | -0.006 |
| | | (-0.950) | (-0.603) |
| SIZE | | | -0.002*** |
| | | | (-11.669) |
| SOE | | | 0.001*** |
| | | | (2.854) |
| AGE | | | -0.002*** |
| | | | (-3.536) |
| LEV | | | 0.000 |
| | | | (0.038) |
| ROA | | | 0.047*** |
| | | | (15.469) |
| FIX | | | -0.006*** |
| | | | (-6.561) |
| INV | | | -0.002 |
| | | | (-0.301) |
| SHR1 | | | -0.000*** |
| | | | (-5.059) |
| Constant | 0.023*** | -0.028 | 0.002 |
| | (91.545) | (-0.369) | (0.032) |
| Province fixed effects | Yes | Yes | Yes |
| Industry fixed effects | Yes | Yes | Yes |
| Year fixed effects | Yes | Yes | Yes |
| Observations | 14314 | 14314 | 14314 |
| Adj_$R^2$ | 0.280 | 0.280 | 0.318 |

Notes

\***, \**, and \* are indicate significant levels of 1%, 5%, and 10% levels, respectively. T-statistics are provided in parentheses below each coefficient estimate.

## 5. Empirical results

### 5.1 Baseline regression

We start our empirical analysis by testing whether the LGD affected corporate R&D investment. Table 3 shows the regression results of Eq (1) using the panel fixed-effect model. To test the robustness of the regression results, the variables are checked step by step. In column (1), we control the province fixed effect, industry fixed effect, and year fixed effect without other control variables. In column (2), we control some provincial-level variables based on the former. All the control variables are included in column (3). The results show that the coefficients of LGD are significantly negative, thus suggesting that LGD will crowd out corporate R&D investment. It verifies Hypothesis 1. In terms of economic significance, for every one standard deviation increase in local government debt, the proportion of corporate R&D investment decreases by 0.037 (-0.002*0.4036/0.022) relative to the mean. In this study, the standard error

is clustered in the province-year two dimensions, and in the robustness check, we cluster standard errors in the province or firm dimension.

## 5.2 Endogenous test in instrumental variables

We use the instrumental variable method to further alleviate potential endogeneity problems. Accordingly, we select the sum of the maturity of all $t-1$ Municipal Bonds as the instrument variable. From a correlation perspective, LGD in year $t-1$ clearly contains the face value of the maturing Municipal Bonds in maturity year $t-1$ (e.g., an LGFV A issued a five-year 100 billion yuan bond in 2008 and issued a three-year 150 billion yuan bond in 2010. In 2012, the LGFV's short-term liability contained 250 billion yuan (100+150) and bank debt). Further, we exclude bonds with a maturity of less than one year to avoid the related bond interference of debt redeemed in the current year. From an exogeneity perspective, the maturity amount of the $t-1$ Municipal Bonds is determined by the time of bond issuance; hence, the exogeneity of the maturity amount of the $t-1$ Municipal Bonds is also satisfied.

The two-step estimation results are presented in Table 4. In column (1), the coefficient of IV is significantly positive in the first IV regression, thereby suggesting that the maturity period of the amount of Municipal Bonds will be reflected in the LGFVs' short-term liabilities. In column (3), the coefficient of LGD is significantly negative in the second regression, which indicates that LGD still crowd out corporate R&D investment after considering the endogenous problems in our model. Meanwhile, the $F$ value tested by Cragg-Donald Wald is 2098.92, which is much higher than the corresponding critical value of stock-Yogo [50]. Therefore, the instrumental variable used in this study is valid.

However, whether the instrumental variable can alleviate the endogeneity of the model depends on the exogeneity of the instrumental variables. The following three methods are used to demonstrate the credibility of the instrumental variable in this study.

**Table 4. Endogenous Test: IV regression.**

| Variables | LGD | | RD | | | |
|---|---|---|---|---|---|---|
| | IV First Stage | | IV Second Stage | | | |
| | (1) | (2) | (3) | (4) | (5) | (6) |
| IV | 0.0002*** | 0.0002*** | | | -0.000** | -0.000 |
| | (2.831) | (2.652) | | | (-2.102) | (-0.781) |
| IV2 | | -0.041** | | | | |
| | | (-2.058) | | | | |
| LGD | | | -0.006*** | -0.006*** | | -0.002*** |
| | | | (-4.175) | (-4.203) | | (-3.524) |
| Other controls | Yes | Yes | Yes | Yes | Yes | Yes |
| Province fixed effects | Yes | Yes | Yes | Yes | Yes | Yes |
| Industry fixed effects | Yes | Yes | Yes | Yes | Yes | Yes |
| Year fixed effects | Yes | Yes | Yes | Yes | Yes | Yes |
| Observations | 13914 | 13914 | 13849 | 13849 | 13848 | 13848 |
| Adj_$R^2$ | 0.952 | 0.952 | 0.316 | 0.316 | 0.317 | 0.317 |
| Cragg-Donald Wald F statistic | 2098.92 | | | | | |
| Sargan Test | | | | 0.804 | | |
| | | | | (0.370) | | |

Notes

\*\*\*, \*\*, and \* are indicate significant levels of 1%, 5%, and 10% levels, respectively. T-statistics are provided in parentheses below each coefficient estimate. Other controls include LnGDP, LnPop, SIZE, SOE, AGE, LEV, ROA, FIX, INV, SHR1.

First, we take both the LGD and IV as explanatory variables simultaneously and perform regressions on corporate R&D investment. If the instrumental variable indirectly affects corporate R&D investment only through the local government debt, then, in this regression, the coefficient of the instrumental variable should not be significant when the LGD is controlled. As presented in column (6) of Table 4, the coefficient of IV is not significant, while the LGD remains significant, which is in line with our expectations. Additionally, when the corporate's R&D investment regresses to the two separately, the coefficients of both are significant. This result implies that the instrumental variable does not directly affect corporate R&D expenditures but only through LGD.

Second, the identification method provided by Larcker and Rusticus [51] is used to determine whether the instrumental variable satisfies the exogenous hypothesis. Larcker and Rusticus [51] (page 191) provide a method to identify whether instrumental variables can still help alleviate the endogeneity problem when it does not sufficiently satisfy the exogeneity assumption (semi-endogenous). Certainly, it is essentially a qualitative identification method. The identification logic is briefly described as follows. Eqs (2) and (3) are the estimators of the beta value under the OLS and IV methods, respectively. Eq (4) is calculated from Ee (2) < Eq (3), and Eq (4) can be converted to express the regression output, that is, Eq (5).

$$plim\ beta_{OLS} = \beta + \frac{cov(x, u)}{var(x)} = \beta + \frac{\sigma_u}{\sigma_x} corr(x, u) \qquad (2)$$

$$plim\ beta_{IV} = \beta + \frac{cov(x, u)}{cov(x, z)} = \beta + \frac{\sigma_u}{\sigma_x} \frac{corr(z, u)}{corr(x, z)} \qquad (3)$$

$$|corr(z, u)| < |corr(x, z)||corr(x, u)| \qquad (4)$$

$$R^2_{zu} < R^2_{xz} R^2_{xu} \qquad (5)$$

where $x$ is the core explanatory variable, $u$ is the residual term, $z$ is the instrumental variable, and $R^2_{zu}$ is the R-squared of the IV one-stage regression. In Eq (4), even if the correlation coefficient between the instrumental variable $z$ and the residual term $u$ is not 0 (i.e., the exogeneity assumption is not fully satisfied), provided that the correlation between the instrumental variable $z$ and the independent variable $x$ is sufficiently strong, we may obtain a better IV estimate result than OLS. It should be noted that we cannot obtain the true residuals. Hence, $corr(z,u)$ and $corr(x,u)$ can not be accurately calculated. As shown in column (1) of Table 4, the R-squared for the first stage of IV is 0.952. Moreover, we directly use the estimated residual (rather than the true residual) to calculate $corr(z,u) = 0.048$. The gap between them is so large that the slight correlation between $x$ and $u$ can meet the requirements of Eq (5). Therefore, we believe that better results than OLS can be obtained, although this instrumental variable may not fully meet the exogenous requirement.

Third, referring to Demirci et al. [7], on the basis of the original IV, we add provincial defense expenditure (IV2) as another instrumental variable of LGD. In terms of relevance, under the premise of limited fiscal expenditure, the higher the defense expenditure, the smaller the local government's expenditure for debt. Further, provincial defense expenditures are relatively exogenous to local economic activities. These two instrumental variables are used to further alleviate the endogeneity caused by reverse causality. The empirical results are presented in columns (2) and (4) of Table 4. It can be observed that the value of the Sargan test cannot reject the null hypothesis; that is, both instrumental variables are exogenous.

# 6. Further analysis

## 6.1 Mechanism analysis

In the theoretical hypothesis, we analyze the three channels through which LGD affects corporate R&D investment, by consuming regional fiscal resources, strengthening government tax collection efforts, and consuming regional credit resources. Now, we further test whether this mechanism exists by introducing three interaction terms in the benchmark regression: the interaction item between LGD and fiscal subsidies, that between LGD and corporate actual tax burden, and that between LGD and external financing dependence. The models are expressed as follows:

$$RD_{p,i,t} = \alpha_1 + \beta_1 LGD_{p,t-1} + \lambda_1 Sub_{p,i,t} + \chi_1 LGD_{p,t-1} * Sub_{p,i,t} + \gamma_1 Control_{p,i,t} + \delta_t + \theta_p + \tau_{ind} + \varepsilon_{p,i,t} \quad (6)$$

$$RD_{p,i,t} = \alpha_2 + \beta_2 LGD_{p,t-1} + \lambda_2 ERT_{p,i,t} + \chi_2 LGD_{p,t-1} * ERT_{p,i,t} + \gamma_2 Control_{p,i,t} + \delta_t + \theta_p + \tau_{ind} + \varepsilon_{p,i,t} \quad (7)$$

$$RD_{p,i,t} = \alpha_3 + \beta_3 LGD_{p,t-1} + \lambda_3 EFD_{p,i,t} + \chi_3 LGD_{p,t-1} * EFD_{p,i,t} + \gamma_3 Control_{p,i,t} + \delta_t + \theta_p + \tau_{ind} + \varepsilon_{p,i,t} \quad (8)$$

where *Sub* is the fiscal subsidy, measured by the logarithm of corporate fiscal subsidies. Further, *ERT* is the corporate actual tax burden, following Hanlon and Heitzman [52], measured by $\frac{income\ tax\ expense}{pre\_tax\ accounting\ profit-deferred\ income\ tax\ expense/nominal\ tax\ rate}$. *EFD* is a dummy variable, representing the enterprise's external financing dependence. Specifically, $EFD = \frac{CE-Adj\_CF}{CE}$, where *CE* is measured by the cash paid by corporates to purchase and construct fixed, intangible, and other long-term assets: *Adj_CF* = operating cash flow + decrease in inventory + decrease in accounts receivable + increase in accounts payable. Finally, we use the median level of all enterprises in each industry to measure the external financing dependency of the industry that year. A corporate is defined as a high external financing dependence, which is higher than the median; otherwise, it is a low external financing dependence [53]. The definitions of other variables are the same as in the baseline regression. If the coefficients of the interaction terms $\chi_1$, $\chi_2$, and $\chi_3$ are statistically significant, then LGD depends on Sub, ERT, and EFD, respectively, to have an impact on corporate R&D investment.

Table 5 reports the regression results of this mechanism test. The explained variables in columns (1), (2), and (3) in Table 5 are corporate R&D investment. The coefficients of LGD in these three columns are all significantly negative. This result indicates that, considering other conditions unchanged, LGD will reduce the corporate R&D investment, which again verifies these conclusions. In column (1) of Table 5, the coefficient of the interaction term between LGD and fiscal subsidies is significantly positive, which implies that fiscal subsidies can alleviate the negative effect of LGD on corporate R&D investment. In column (2), the coefficient of the interaction term between LGD and the corporate actual tax burden is significantly negative. This fact indicates that LGD inhibits the corporate R&D investment by increasing the corporate actual tax burden. In column (3), the coefficient of the interaction term between LGD and external financing dependence is significantly negative. This result indicates that LGD inhibits enterprise's R&D expenditures by reducing external financing of enterprises.

In summary, LGD affects corporate R&D investment by consuming regional fiscal resources, strengthening government tax collection efforts, and consuming regional credit resources.

## 6.2 Cross-sectional analysis

**6.2.1 Cross-sectional analysis (SOE/NSOE).** The SOEs have natural political connections and comparative advantages over NSOEs in terms of fiscal resources, tax collection efforts, and credit resources, which cause NSOEs to be more sensitive to the government's behavior. Therefore, in this section, the samples are divided into SOE and NSOE according to corporate

**Table 5. Mechanism analysis.**

| Variables | Dependent variable: RD | | |
|---|---|---|---|
| | **(1)** | **(2)** | **(3)** |
| LGD | -0.007** | -0.001** | -0.002** |
| | (-2.476) | (-2.100) | (-2.547) |
| Sub | 0.002*** | | |
| | (8.080) | | |
| ERT | | -0.005*** | |
| | | (-5.274) | |
| RZ | | | -0.001 |
| | | | (-1.552) |
| LGD*Sub | 0.0003* | | |
| | (1.951) | | |
| LGD*ERT | | -0.002** | |
| | | (-2.070) | |
| LGD*RZ | | | -0.002* |
| | | | (-1.907) |
| Other controls | Yes | Yes | Yes |
| Province fixed effects | Yes | Yes | Yes |
| Industry fixed effects | Yes | Yes | Yes |
| Year fixed effects | Yes | Yes | Yes |
| Observations | 11980 | 14133 | 14314 |
| Adj_$R^2$ | 0.339 | 0.329 | 0.318 |

Notes

***, **, and * are indicate significant levels of 1%, 5%, and 10% levels, respectively. T-statistics are provided in parentheses below each coefficient estimate. Other controls include LnGDP, LnPop, SIZE, SOE, AGE, LEV, ROA, FIX, INV, and SHR1.

property rights, and the two samples are regressed based on model (1). The empirical results are listed in Table 6. In columns (1) and (2), the coefficient of LGD is negative and significant only in NSOE samples. This result suggests that LGD will crowd out corporate R&D investment when the firm is NSOE. Therefore, the crowding-out effect of LGD on R&D investment is more significant for NSOEs.

**6.2.2 Cross-sectional analysis (large and small).** In this section, the samples are divided into large-size and small-size enterprises according to corporate assets, and the two samples are regressed based on model (1). The empirical results are shown in columns (3) and (4) of Table 6. We can observe that the coefficient of LGD is negative and only significant in small-size enterprise samples. This result indicates that LGD will crowd out corporate R&D investment when the firm is small. It may because the innovation of small-scale enterprises is more dependent on fiscal resources from the local government and bank credit resources. When an increase in LGD leads to a reduction in fiscal resources and credit resources, small-scale enterprises will naturally be more sensitive. Therefore, the crowding-out effect of LGD on corporate R&D investment is more significant for small enterprises.

**6.2.3 Cross-sectional analysis (old and young).** The old enterprise may already have a certain degree of self-innovation ability in the development process, and simultaneously, it accumulates more resources, such as capital and knowledge needed for innovation. However, young enterprises may have insufficient accumulation of innovation resources due to their short development time. Their innovative activities must rely more on the support of external

**Table 6. Cross-sectional analysis 1.**

| Variables | Dependent variable: RD | | | |
|---|---|---|---|---|
| | Property Rights | | Size | |
| | NSOE | SOE | Small | Large |
| | (1) | (2) | (3) | (4) |
| LGD | -0.002** | -0.001 | -0.003*** | -0.001 |
| | (-2.349) | (-1.631) | (-3.465) | (-1.529) |
| Other controls | Yes | Yes | Yes | Yes |
| Province fixed effects | Yes | Yes | Yes | Yes |
| Industry fixed effects | Yes | Yes | Yes | Yes |
| Year fixed effects | Yes | Yes | Yes | Yes |
| Observations | 10230 | 4083 | 7163 | 7151 |
| Adj_$R^2$ | 0.298 | 0.362 | 0.284 | 0.326 |

Notes

***, **, and * are indicate significant levels of 1%, 5%, and 10% levels, respectively. T-statistics are provided in parentheses below each coefficient estimate. Other controls include LnGDP, LnPop, SIZE, SOE, AGE, LEV, ROA, FIX, INV, SHR1.

resources. Therefore, in this section, the samples are divided into old and young enterprises, according to the length of time that firms have been established in the region, and the two samples are regressed based on Model (1). The empirical results are listed in columns (1) and (2) in Table 7. It can be observed that the coefficient of LGD is negative and only significant in young enterprise samples, thus implying that the local government's implicit debt will crowd out R&D investment in younger firm.

**6.2.4 Cross-sectional analysis (industry competition).** In this section, the samples are divided into high-industry competitive enterprises and low-industry competitive enterprises according to industry concentration (the proportion of enterprise assets in the industry), and two samples are regressed based on model (1). The empirical results are listed in Table 7. In columns (3) and (4), we observe that the coefficient of LGD is negative and significant only in low-industry competitive enterprise samples, which suggests that LGD will crowd out R&D investment when the firm low-industry competitive. This result is because in highly

**Table 7. Cross-sectional analysis 2.**

| Variables | Dependent variable: RD | | | |
|---|---|---|---|---|
| | Firm Age | | Industry Competition | |
| | Old | Young | High | Low |
| | (1) | (2) | (3) | (4) |
| LGD | -0.001 | -0.003*** | -0.000 | -0.003*** |
| | (-1.558) | (-3.072) | (-0.108) | (-4.253) |
| Other controls | Yes | Yes | Yes | Yes |
| Province fixed effects | Yes | Yes | Yes | Yes |
| Industry fixed effects | Yes | Yes | Yes | Yes |
| Year fixed effects | Yes | Yes | Yes | Yes |
| Observations | 6432 | 7882 | 3155 | 11159 |
| Adj_$R^2$ | 0.325 | 0.319 | 0.232 | 0.339 |

Notes

***, **, and * are indicate significant levels of 1%, 5%, and 10% levels, respectively. T-statistics are provided in parentheses below each coefficient estimate. Other controls include LnGDP, LnPop, SIZE, SOE, AGE, LEV, ROA, FIX, INV, and SHR1.

fragmented industries (high industry), even if a large amount of LGD brings about changes in local government behavior, its impact (externality) on this industry is very limited. However, in the low-industry, the performance of each firm is highly dependent on the choices of other firms in the industry. In this case, the R&D investment behavior of non-urban investment corporations (belonging to low industry) is more easily affected by the behavior of local governments. Therefore, the crowding-out effect of LGD on R&D investment is more significant for low-industry competitive enterprises.

## 7. Robustness checks

### 7.1 Alternative measures

The baseline results may be sensitive to different definitions of the key variables. To determine whether the measurements of RD and LGD are robust. On the one hand, we employ three alternative measures of corporate R&D investment as the dependent variable: (1) RD2 (RD2 = Log (R&D investment+1)); (2) RD3 (RD3 = R&D investment/operating income); (3) RD4 (RD4 = R&D investment/operating income). Columns (1), (2), and (3) in Table 8 presents the estimation results. We can see that the coefficients of all LGD remain significant negative, respectively. This result supports our main finding that the local government implicit debt will crowd out corporate R&D investment, even when we take the three alternative measures of corporate R&D investment.

On the other hand, we take LGD2 (LGD2 = (short-term borrowing + notes payable + one-year current liabilities + other current liabilities + short-term bonds payable)/fiscal revenue) as alternative measures of the independent variable. The estimation results are shown in column (4) of Table 8. The coefficient of LGD2 is significantly negative, which supports our conclusion that LGD will crowd out corporate R&D investment.

### 7.2 Excluding other fiscal and taxation policies interference

There were some fiscal and taxation policies in China during 2012–2018, such as the value-added tax (VAT) policy in 2012 and the depreciation of fixed assets pilot policy (DFAP) in

**Table 8. Robust Test 1.**

| Variables | RD2 | RD3 | RD4 | RD | | |
|---|---|---|---|---|---|---|
| | (1) | (2) | (3) | (4) | (5) | (6) |
| LGD | -0.145*** | -0.002* | -0.016** | | -0.002*** | -0.001*** |
| | (-2.623) | (-1.875) | (-2.545) | | (-3.921) | (-2.691) |
| LGD2 | | | | -0.001** | | |
| | | | | (-2.486) | | |
| Other controls | Yes | Yes | Yes | Yes | Yes | Yes |
| Province fixed effects | Yes | Yes | Yes | Yes | Yes | No |
| Industry fixed effects | Yes | Yes | Yes | Yes | Yes | No |
| Year fixed effects | Yes | Yes | Yes | Yes | Yes | Yes |
| Industry*year fixed effects | No | No | No | No | Yes | No |
| Firm fixed effect | No | No | No | No | No | Yes |
| Observations | 14314 | 14314 | 9251 | 14314 | 14305 | 14099 |
| Adj_$R^2$ | 0.564 | 0.383 | 0.385 | 0.318 | 0.316 | 0.848 |

Notes

***, **, and * are indicate significant levels of 1%, 5%, and 10% levels, respectively. T-statistics are provided in parentheses below each coefficient estimate. Other controls include LnGDP, LnPop, SIZE, SOE, AGE, LEV, ROA, FIX, INV, SHR1.

2014, which would influence the estimation results of our main findings. As most of these policies are based on industry, we hope to control the influence of policies by controlling the industry-year fixed effect. Table 8 presents the estimation results. In column (5), we can observe that the coefficient of LGD is still significantly negative, indicating that the estimated results of this paper are relatively robust.

In the benchmark model, we control province, industry, and time fixed effects to eliminate the influence of some omitted variables that do not vary with the province, industry, and time on the estimated results. Furthermore, to eliminate the influence of omitted variables that do not change with enterprises or time on the estimation, we control firm fixed effects and year fixed effects. The results are shown in Column (6) of Table 8; the coefficient of LGD remains significantly negative, thus suggesting that LGD will crowd out corporate R&D investment.

### 7.3 Changing the empirical model

We add a one-period lagging variable of the dependent variable corporate R&D investment ($RD_{p,i,t-1}$) on the basis of the original benchmark model, as expressed in formula (9). The current empirical model is a dynamic panel model, and the system GMM is employed to estimate formula (9) to ensure the accuracy of the results.

$$RD_{p,i,t} = \alpha_4 + \beta_4 LGD_{p,t-1} + \sigma_4 RD_{p,i,t-1} + \gamma_4 Control_{p,i,t} + \delta_t + \theta_p + \tau_{ind} + \varepsilon_{p,i,t} \tag{9}$$

The estimation results are presented in column (1) of Table 9. We observe that the

**Table 9. The estimation results of the system GMM.**

| Variables | Dependent variable: RD | | |
|---|---|---|---|
| | SYS-GMM | SYS-GMM | SYS-GMM |
| | (1) | (2) | (3) |
| LGD | -0.003** | -0.002** | -0.002** |
| | (-2.405) | (-2.367) | (-2.367) |
| L.RD | 0.510*** | 0.462*** | 0.439** |
| | (7.831) | (4.602) | (2.424) |
| L2.RD | | 0.158* | 0.123* |
| | | (1.824) | (1.845) |
| L3.RD | | | 0.039 |
| | | | (0.615) |
| Other controls | Yes | Yes | Yes |
| Province fixed effects | Yes | Yes | Yes |
| Industry fixed effects | Yes | Yes | Yes |
| Year fixed effects | Yes | Yes | Yes |
| Observations | 10950 | 8024 | 5742 |
| AR(1) | -7.613*** | -5.742*** | -5.041*** |
| | 0.000 | 0.000 | 0.000 |
| AR(2) | -0.441 | 1.173 | 3.180 |
| | 0.660 | 0.242 | 0.520 |
| Hansen test | 5.81 | 7.93 | 11.82 |
| | 0.214 | 0.474 | 0.365 |

Notes

***, **, and * are indicate significant levels of 1%, 5%, and 10% levels, respectively. T-statistics are provided in parentheses below each coefficient estimate. Other controls include LnGDP, LnPop, SIZE, SOE, AGE, LEV, ROA, FIX, INV, and SHR1.

coefficient of the one-period lagging variable of corporate R&D investment ($RD_{p,i,t-1}$) is significantly positive and that of the LGD is significantly negative. This result implies that previous corporate R&D investment has a positive impact on current R&D activities. However, it does not affect the main conclusion of this study: LGD reduces corporate R&D investment. Further, to ensure the robustness of the results, we successively add two-period and three-period lagging variables of corporate R&D investment in formula (9). The results are shown in columns (2) and (3) of Table 9. It can be observed that the influence of the corporate past R&D investment on the current decision-making gradually weakens. The three-period lagging variables of corporate R&D investment in column (3) has no significant impact on the current R&D decision-making. At the same time, in columns (2) and (3) of Table 9, the coefficients of LGD have remained significantly negative, thus indicating that the LGD has reduced the corporate R&D investment. This finding is consistent with these conclusions and ensures the robustness of the benchmark estimation results.

The validity of the system GMM estimation results must meet two conditions: the random error term does not have serial correlation and the weak instrumental variable problem does not exist. The corresponding AR (1), AR (2), and Hansen statistics are shown in columns (1), (2), and (3) of Table 9. Clearly, AR (1) is significant at the 1% level, and AR (2) is not significant, thereby implying that no serial correlation exists in the random error term in the model. The Hansen statistics show that the null hypothesis cannot be rejected, thus implying that the weak instrumental variable problem does not exist in models. Therefore, these results are valid.

### 7.4 Using province or enterprise dimensions cluster standard errors

In the benchmark model, we cluster the standard error to the double dimension of province-year. To show that standard errors do not affect the benchmark results of this study, we use the province dimension and enterprise dimension cluster standard errors, respectively. The empirical results are presented in columns (1) and (2) of Table 10. The coefficients of LGD are

**Table 10. Robust Test 2.**

| Variables | Dependent variable: RD | | | | |
|---|---|---|---|---|---|
| | **(1)** | **(2)** | **(3)** | **(4)** | **(5)** |
| LGD | -0.002** | -0.002** | -0.002*** | -0.002*** | -0.001** |
| | (-2.372) | (-2.565) | (-4.072) | (-3.396) | (-2.181) |
| TLR | | | -0.009* | | |
| | | | (-1.837) | | |
| LGB | | | 0.001 | | |
| | | | (0.198) | | |
| CB | | | -0.004 | | |
| | | | (-0.353) | | |
| Other controls | Yes | Yes | Yes | Yes | Yes |
| Province fixed effects | Yes | Yes | Yes | Yes | Yes |
| Industry fixed effects | Yes | Yes | Yes | Yes | Yes |
| Year fixed effects | Yes | Yes | Yes | Yes | Yes |
| Observations | 14314 | 14314 | 14303 | 11617 | 14314 |
| Adj_$R^2$ | 0.318 | 0.318 | 0.318 | 0.294 | 0.214 |

Notes

***, **, and * are indicate significant levels of 1%, 5%, and 10% levels, respectively. T-statistics are provided in parentheses below each coefficient estimate. Other controls include LnGDP, LnPop, SIZE, SOE, AGE, LEV, ROA, FIX, INV, and SHR1.

still significantly negative, thereby suggesting that LGD will crowd out corporate R&D investment even when we use the different dimensions of cluster standard errors.

## 7.5 Controlling for other provincial variables

To control the interference of potential provincial variables on the estimated results, we added other provincial control variables such as the ratio of the transferred land revenue to fiscal expenditure (TLR), the ratio of the issuance amount of local government bonds to fiscal expenditure (LGB), and the ratio of the issuance amount of Municipal Bonds to fiscal expenditure (CB). The empirical results presented in column (3) of Table 10 show that the coefficient of LGD is significantly negative, thus suggesting that LGD will crowd out corporate R&D investment even when we control for other provincial control variables.

## 7.6 Excluding four municipalities directly under the Central Government

Four municipalities Beijing, Tianjin, Shanghai, and Chongqing have a special political level in China, where the innovation activities in this region may be affected. Therefore, we exclude the enterprises belonging to these four municipalities. The empirical results are presented in Table 10. In column (4), we show that the coefficient of LGD is still significantly negative, which indicates that LGD will crowd out corporate R&D investment even when we consider special samples.

## 7.7 Using original data

In the baseline regression, we adopt the commonly used treatment of extreme values in the current literature, in which all continuous variables are winsorized at the top and bottom 1% to avoid the impact of outliers. Considering that the sample of extreme values may contain additional information, these variations may affect the estimated results. Thus, the original data are adopted in this section. The empirical results are shown in Table 10. In Column (5), the coefficient of LGD is negative, thereby suggesting that LGD will crowd out corporate R&D investment even when we consider the influence of the extreme values of the variables on the estimated results.

## 8. Conclusions

Based on the data of Chinese listed companies and LGFVs from 2012 to 2018, we empirically test the impact of LGD on corporate R&D investment from the perspective of LGD redemption. We find that LGD will crowd out R&D investments. In terms of economic significance, with every standard deviation increase in local government debt, the proportion of corporate R&D investment decreases by 0.037. The crowding-out effect is more pronounced when the firm is NSOE, when the firm's size is small, or when the firm is in lower market competition. Overall, our findings indicate that LGD crowds out corporate R&D investment. We further provide suggestive evidence that consuming fiscal resources, strengthening tax collection efforts, or consuming credit resources might partly account for the crowding-out effect.

Our empirical findings provide some policy implications. First, the government should make the market-oriented transformation for local government financing platform companies. The prerequisite for the transformation is to strip off the platform's government financing function and properly handle the platform's existing debt. On this basis, the local government can revitalize the city's stock assets to make LGFVs' have hematopoietic functions and be a real entity enterprise, ultimately avoiding its distortion of resources for non-urban investment companies. Second, the government should comply with the law of market optimal

resource allocation and weaken its preference for special enterprises. Third, the government must relax credit constraints or give tax incentives to innovative enterprises. Innovation is a high-risk and long-term investment; hence, a stable cash flow in the early stage is required to ensure the progress of research and development. Relaxing credit constraints or giving tax incentives will motivate companies to better invest in R&D.

Further research can be undertaken in the following directions. First, the measurement of LGD in this study is calculated from LGFVs that have issued bonds. However, there are LGFVs that have not issued bonds; researchers could get the debt situation from other ways. Second, to eliminate the crowding-out effect of LGD on corporate R&D investment. We use listed firms; one could test whether the results also hold for unlisted firms. Third, making a distinction between "research" (R) and "development" (D) has been emphasized by some studies. However, the specific information contained in the data from listed firms in China does not support us from doing this type of exercise.

## Author Contributions

**Conceptualization:** Junbing Xu.

**Data curation:** Junbing Xu, Yang He.

**Formal analysis:** Junbing Xu, Dawei Feng, Yang He.

**Funding acquisition:** Yuanyuan Li.

**Methodology:** Dawei Feng, Yang He.

**Supervision:** Yuanyuan Li.

**Validation:** Dawei Feng.

**Writing – original draft:** Junbing Xu.

**Writing – review & editing:** Yuanyuan Li, Dawei Feng, Zhouyi Wu, Yang He.

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
