## [Decision Letter · Decision Letter 0]

14 Jul 2021

PONE-D-21-10527

Crowding in or crowding out? How local government debt influence corporate innovation for China

PLOS ONE

Dear Dr. Feng,

Thank you for submitting your manuscript to PLOS ONE. After careful consideration, we feel that it has merit but does not fully meet PLOS ONE’s publication criteria as it currently stands. Therefore, we invite you to submit a revised version of the manuscript that addresses the points raised during the review process.

We look forward to receiving your revised manuscript.

Kind regards,

Elisa Ughetto

Academic Editor

PLOS ONE

A clean copy of the edited manuscript (uploaded as the new *manuscript* file).

Additional Editor Comments (if provided):

Reviewers' comments:

Reviewer's Responses to Questions

**Comments to the Author**

1. Is the manuscript technically sound, and do the data support the conclusions?

Reviewer #1: Partly

2. Has the statistical analysis been performed appropriately and rigorously? 

Reviewer #1: No

3. Have the authors made all data underlying the findings in their manuscript fully available?

Reviewer #1: Yes

4. Is the manuscript presented in an intelligible fashion and written in standard English?

Reviewer #1: No

5. Review Comments to the Author

Reviewer #1: This study examines the effects of local government debt on corporate R&D activities by focusing on the case of China. The authors find the crowding-out effects, and they argue that three channels (consumption of fiscal resources, strengthening tax collection efforts, consumption of credit resources) are channels that lead to these results. While the study tackles the interesting question with a valid empirical framework, I believe that there is much room for the improvement of the current document. Here, I mainly provide four suggestions.

For more information, plaese find the attached file.

6. PLOS authors have the option to publish the peer review history of their article (what does this mean?). If published, this will include your full peer review and any attached files.

Reviewer #1: No

---

## [Author Response · Author response to Decision Letter 0]

19 Aug 2021

Regarding the response to reviewers, please click on the attachment.

---

## [Editor Report · Decision Letter 1]

20 Oct 2021

Crowding in or crowding out? How local government debt influences corporate innovation for China

PONE-D-21-10527R1

Dear Dr. Feng,

We’re pleased to inform you that your manuscript has been judged scientifically suitable for publication and will be formally accepted for publication once it meets all outstanding technical requirements.

Kind regards,

Elisa Ughetto

Academic Editor

PLOS ONE

Additional Editor Comments (optional):

The paper has been much improved from the last version. I am pleased to accept it.
---

## [Editor Report · Acceptance letter]

29 Oct 2021

PONE-D-21-10527R1 

Crowding in or crowding out? How local government debt influences corporate innovation for China 

Dear Dr. Feng:

I'm pleased to inform you that your manuscript has been deemed suitable for publication in PLOS ONE. Congratulations! Your manuscript is now with our production department. 

Kind regards, 

on behalf of

Prof. Elisa Ughetto 

Academic Editor

PLOS ONE